# Dietary diversity and nutritional status of adults living with HIV during the COVID-19 era

**Kasim Abdulai**[ID][1]*, **Abdul Rauf Alhassan**[ID][2], **Safianu Osman Aleboko**[1], **Mohammed Doobia Ahmed**[3], **Awal Seidu Mohammed**[1], **Odei-Asare Fremah Adom**[1], **Rhoda Kumah**[1]

1 Department of Nutrition and Dietetics, University of Cape Coast, Cape Coast, Ghana, 2 Department of Surgery, Tamale Teaching Hospital, Tamale, Ghana, 3 Gushegu Midwifery Training College, Gushegu, Ghana

* Kasim.abdulai@ucc.edu.gh

## Abstract

### Background

The coronavirus Disease 2019 (COVID-19) pandemic has brought about unique challenges in healthcare and nutrition, particularly for people living with HIV (PLHIV). Understanding their dietary patterns and nutritional status is crucial for developing targeted interventions and improving health outcomes. Therefore, this study assessed the dietary diversity and nutritional status of PLHIV during the COVID-19 era.

### Methods

We adopted a facility-based cross-sectional study design to enroll 220 PLHIV from two hospitals in the Central Region of Ghana. Dietary intakes were assessed using 24-hour recall. Anthropometric and body composition data were collected with a stadiometer and a body composition monitor. Dietary diversity was evaluated using the FAO's Individual Dietary Diversity Score (IDDS). Data analysis was conducted with SPSS version 20. Significance level was set p-value less than 0.05.

### Results

A significant proportion (33.2%) of PLHIV had low dietary diversity, with the majority (55.5%) categorized as needing dietary improvement. Approximately 2 out of every 10 of the participants were identified as underweight. Participants aged 40 to 59 years were more likely to exhibit higher dietary diversity (adjusted odds ratio (AOR) = 1.966, 95% Confidence Interval (CI): 1.045–4.987). Participants who consumed meals at least three times daily were more likely to have a high IDDS (AOR = 1.641, 95% CI: 1.221, 8.879). Employed participants (public sector and private sector) were also more likely to have a high IDDS compared to unemployed participants (AOR = 1.448, 95% CI: 1.028–3.042; AOR = 1.165, 95% CI: 1.030–9.329, respectively). Factors associated with undernutrition included being female (AOR = 1.829, 95% CI: 1.294, 3.872) and first-line antiretroviral therapy ART (AOR = 1.683, 95% CI: 1.282–2.424).

**Data Availability Statement:** All relevant data are within the manuscript and its Supporting Information files.

**Funding:** The author(s) received no specific funding for this work.

**Competing interests:** The authors have declared that no competing interests exist.

## Conclusion

The study emphasizes the need for nutritional interventions for PLHIV, particularly during crises. It advocates for a policy collaboration to address food insecurity and promote resilient health outcomes.

## Introduction

The global human immunodeficiency virus/acquire immunodeficiency syndrome (HIV/AIDS) pandemic has stood as one of the most formidable public health challenges in recent history. Despite remarkable strides in treatment and care, HIV/AIDS continues to cast its shadow upon millions of individuals across the globe. By the close of 2022, approximately 39 million people were grappling with HIV worldwide [1]. Notably, the World Health Organization (WHO) African Region, encompassing Ghana, bore the brunt of this epidemic, harboring two-thirds of this afflicted population, equating to 25.6 million individuals. Regrettably, during this same period, 630,000 lives were tragically lost to HIV-related causes, and a staggering 1.3 million new HIV infections were reported [1].

In recent times, the emergence of the COVID-19 pandemic has ushered in a new layer of complexity within the healthcare landscape, significantly impacting vulnerable communities, including people living with HIV (PLHIV). The confluence of these two pandemics has ushered forth pressing questions concerning the dietary diversity and nutritional well-being of PLHIV in the era of COVID-19 [2]. Malnutrition poses a substantial hurdle for PLHIV, as it undermines their immune systems and heightens susceptibility to opportunistic infections [2].

Nutrition occupies a central role in the lives of PLHIV, exerting a profound influence on their health and overall quality of life [3]. Maintaining an optimal diet is crucial for bolstering the immune system, managing the side effects of antiretroviral therapy (ART), and nurturing general health [4]. PLHIV frequently confront a heightened risk of malnutrition due to factors such as increased nutritional demands, diminished appetite, gastrointestinal issues, and co-existing health conditions [5]. Upholding a varied and balanced diet is imperative for PLHIV, ensuring they receive the requisite nutrients to reinforce their immune defenses, alleviate symptoms, and curtail the risk of opportunistic infections.

Clinical investigations have revealed that PLHIV exhibit a heightened incidence of diarrhea, which leads to malabsorption and nutrient losses [6]. Studies have also unearthed a higher prevalence of deficiencies in essential micronutrients such as vitamin A, vitamin B12, vitamin C, vitamin D, selenium, zinc, and iron among PLHIV [7]. Notably, deficiencies in vitamin B12, zinc, and selenium have been correlated with compromised immune function and an elevated risk of disease progression in PLHIV [2]. The significance of dietary diversity is accentuated by the challenges ushered in by the COVID-19 pandemic, including disruptions in food supply chains, economic hardship, and restricted access to healthcare services.

The ramifications of the COVID-19 pandemic have resonated deeply within communities and individuals across the globe. For PLHIV, this crisis has posed unique challenges, intersecting with their HIV status. The term "COVID-19 era" typically refers to the period of time marked by the emergence and global spread of the COVID-19 pandemic caused by the novel coronavirus Severe Acute Respiratory Syndrome Coronavirus 2 (SARS-CoV-2) [8]. This era began in late 2019 when the virus was first identified in Wuhan, China, and has extended into subsequent years as the world continues to grapple with the pandemic. The specific timeframe can vary depending on the context and region, but it generally encompasses the years from

2019 to the present and possibly into the foreseeable future as efforts to control and manage the pandemic continue.

Lockdowns, mobility restrictions, and disruptions to healthcare services as a result of COVID-19 have impeded PLHIV's access to essential antiretroviral therapy, routine medical check-ups, and nutritional assistance. Moreover, the economic downturn triggered by the pandemic has engendered financial strains, potentially leading to food insecurity and limited availability of nourishing foods. The social isolation and stress induced by the pandemic have also cast a pall over mental health [9], further influencing dietary choices and nutritional well-being among PLHIV. Particularly, immunocompromised individuals face an exceptionally high risk of COVID-19 infection, and malnutrition magnifies this risk manifold.

Given the intricate interplay between HIV/AIDS, nutrition, and the COVID-19 pandemic, a pressing need arose to probe into the dietary diversity and nutritional status of PLHIV during this unprecedented COVID-19 era. Understanding the challenges and opportunities entailed in securing adequate nutrition for PLHIV amid the COVID-19 crisis is pivotal for optimizing their health outcomes and overall well-being. It informs public health interventions, shapes clinical practices, and guides policy decisions aimed at more effectively supporting the nutritional requirements of PLHIV. In developing countries, including Ghana, there is a dearth of comprehensive research addressing this crucial public health concern, particularly during the COVID-19 pandemic. Consequently, this study sought to assess dietary diversity and nutritional status, along with their determinants, among HIV patients attending ART clinics at two public hospitals in the Central Region of Ghana.

## Methods

### Study design and settings

We employed a facility-based cross-sectional study design to conduct our research, which spanned from 15th September, 2021 to 30th to October, 2021. The study focused on PLHIV aged 18 years or older who had been receiving ART for a minimum of six months. Our study was conducted at two healthcare facilities: Cape Coast Teaching Hospital and University of Cape Coast Hospital, both of which offer ART services. These facilities are situated within Cape Coast Metropolis in the Central Region of Ghana. We randomly selected 220 PLHIV who met the inclusion criteria and who consented to participate in our research.

### Sample size determination

The required sample size was calculated using a formula for determining sample size for a single population proportion [10], using 17% [11] as the proportion of undernutrition (P) with a 5% level of significance, at a 95% level of confidence for a two-tail test, and a marginal error or level of precision (d) = 5%. The sample size (n) was determined as follows:

$$n = \frac{Z^2 * P(1-P)}{d^2} = \frac{1.96^2 * 0.17(1-0.17)}{0.05^2} = \mathbf{217}$$

The minimum sample size of 217 was sufficient to answer the research question. However, we rounded it up to 220.

### Sampling procedure

Both the hospitals (study sites) and the study participants were randomly selected. A computerized random selection method in R was used to randomly select the two hospitals. A predefined list of the hospitals in the Cape Coast Metropolis was created, named Hospital A through

Hospital F. A computerized method, with a fixed seed for reproducibility, was then utilized to ensure an unbiased and replicable selection process. This method resulted in the selection of Hospital F and Hospital D, which represented Cape Coast Teaching Hospital (CCTH) and University of Cape Coast Hospital (UCCH) respectively. This approach guarantees fairness in the selection process and supports the scientific integrity of the study by mitigating potential selection biases [12].

A total of 220 participants were recruited from two hospitals: CCTH and UCCH, using a sample allocation method based on probability proportional to size. Given the number of active patients at each hospital (1300 at CCTH and 850 at UCCH), the fraction of the total sample size allocated to each was calculated from their proportion of the combined patient count. This calculation led to 133 participants being selected from CCTH and 87 from UCCH.

Participants were chosen through a random sampling method. Initial observations showed an average of 30 PLHIV attending clinic days at CCTH and about 25 at UCCH. To meet the quota, 15 participants were randomly selected during each ART clinic day at CCTH until the total of 133 was reached; similarly, 10 participants were selected on each clinic day at UCCH until the total of 87 was achieved. On clinic days, potential participants were asked to draw from a box containing slips marked "YES" or "NO," with 15 "YES" slips for CCTH and 10 for UCCH. Those who drew a "YES" slip and provided consent were enrolled in the study.

## Inclusion criteria

The inclusion criteria for PLHIV in this study encompassed several key aspects. Firstly, participants had to meet a minimum age requirement of 18 years. Additionally, eligibility hinged on the confirmation of a sero-positive diagnosis for HIV. Those who had been receiving ART for a duration of at least six months, with their ART treatment administered either at the Cape Coast Teaching Hospital or the University of Cape Coast Hospital, were considered eligible for inclusion. Furthermore, individuals were required to provide informed written consent, thereby demonstrating their voluntary willingness to participate in the research. These comprehensive criteria were employed to ensure a relevant and well-defined participant group for the study.

## Study variables

The primary outcome variables in the study encompassed dietary diversity and nutritional status. The study's explanatory variables included alcohol consumption, type of ART drug used, duration of exposure to ART, smoking habits, age, gender, sex, level of education, and exercise.

## Data collection methods and procedures

Data regarding sociodemographic characteristics, including gender, age, ethnicity, religion, and occupation, was gathered through interviewer administered questionnaire. To assess the dietary intake of participants, a 24-hour recall of their usual food consumption was conducted, in this process, comprehensive details regarding all meals, snacks, and beverages consumed within the previous 24 hours were documented.

## Dietary diversity

As outlined earlier within this paper, individual dietary diversity scores (IDDS) was chosen to assess the dietary diversity of the participants [13–15] of the participants. The diet quality of the study participants was determined using data on their usual food intakes from the

24-Hour Recall. The IDDS was used to determine the diversity of their diet [16–18]. The IDDS used was a modified version of the Food and Agricultural Organization (FAO) dietary diversity questionnaire [14]. The FAO dietary diversity questionnaires is a 12-item scale designed to assess the variety of the diet by summing the number of food groups eaten by household members but uses 9-item scale for individuals in the last 24 hours [14,19]. The 12 major food groups inquired about are vegetables, fruits, cereals, meat, fish, tubers, legumes, eggs, milk and milk products, fats and oils, sugar and sweets, beverages. The reference period can either be the previous day or week [14].

At the household level, the dietary diversity score serves as an indicator of food accessibility, reflecting a household's ability to access various food groups, including those that may be more expensive. On an individual level, IDDS offer straightforward and validated metrics for assessing dietary diversity, quality, and nutrient sufficiency [14]. In this study, the IDDS of PLHIV was derived on the basis of the number of food groups consumed from a 24-hour recall. Any food group consumed in the past 24 hours was given a score of one (1) which was aggregated to give the IDDS, with a maximum possible score of nine (9). According to FAO categorization, a score of zero (0) was assigned to a food category if not consumed in the past 24 hours. A score of 3 or less indicates lower IDDS, a score of 4 and 5 indicates medium IDDS, and a score of 6 or more indicates high IDDS.

## Data management and analysis

Data collected underwent a thorough review and correction process in accordance with a data cleaning protocol. Subsequently, investigators conducted checks for accuracy, consistency, and completeness. The gathered data was then exported to IBM SPSS Statistics 20 for analysis. Descriptive tabulations for key variables were created using univariate analysis. Predictors of the primary outcome, which encompassed dietary diversity and nutritional status, were determined through a multivariable ordinal logistic regression model that incorporated socio-demographic and clinical attributes of PLHIV as potential predictors. Variables with a $p < 0.20$ in the bivariate analysis were selected for inclusion in the multiple regression model. Several factors were considered as potential predictors, and pre-identified confounders including sex, occupation, and exercise status were included. Multivariable ordinal logistic regression modeling, employing the "Enter" method in SPSS, was used to simultaneously enter all variables into a full model generated in a single step, with a significance level of $P < 0.05$ denoting statistical significance.

## Ethical considerations

The study protocol underwent review and received approval from the Cape Coast Teaching Hospital Ethical Review Committee (CCTHERC) with the approval ID: CCTHERC/EC/2021/070, to ensure its compliance with both local and international standards aimed at safeguarding the rights and well-being of human research subjects. A written informed consent was diligently secured from all participants, following a comprehensive explanation of the study's objectives and methodology. Furthermore, participants were provided with assurances of privacy and confidentiality.

## Sociodemographic characteristics and some health-related behaviors of the study participants

Table 1 presents information about the baseline sociodemographic characteristics and health-related behaviors of 220 study participants. The study reveals a diverse demography, with majority (60.5%) of participants falling within the age group of 40–59 years. Again, most of the participants were females (86.8%), with a diverse marital status among the participants. Over

**Table 1. Sociodemographic characteristics and some health-related behaviors of the study participants Top of Form.**

| Factor | Category | Frequency (n) | Percentage (%) |
|---|---|---|---|
| Age Category | 20–39 | 36 | 16.36 |
| | 40–59 | 133 | 60.45 |
| | 60+ | 51 | 23.18 |
| Sex | Female | 191 | 86.82 |
| | Male | 29 | 13.18 |
| Marital Status | Single | 68 | 30.91 |
| | Married | 43 | 19.55 |
| | Divorced | 15 | 6.82 |
| | Widowed | 68 | 30.91 |
| | Separated | 3 | 1.36 |
| | Cohabiting | 23 | 10.45 |
| Level of education | None | 58 | 26.36 |
| | Primary | 43 | 19.55 |
| | Middle/JHS | 94 | 42.73 |
| | Secondary/SHS | 15 | 6.82 |
| | Higher | 10 | 4.55 |
| Occupation | Unemployed | 22 | 10.00 |
| | Farmer | 12 | 5.45 |
| | Artisan | 28 | 12.73 |
| | Public Sector | 16 | 7.27 |
| | Private Sector | 5 | 2.27 |
| | Trading | 135 | 61.36 |
| | Others | 2 | 0.91 |
| Do you smoke cigarette? | Yes | 9 | 4.09 |
| | No | 211 | 95.91 |
| Do you drink alcohol? | Yes | 168 | 76.36 |
| | No | 52 | 23.64 |
| Do you exercise? | Yes | 59 | 26.82 |
| | No | 161 | 73.18 |
| Do you eat outside home? | Yes | 21 | 9.55 |
| | No | 199 | 90.45 |
| How many times do you eat in a day? | Once | 4 | 1.82 |
| | Twice | 55 | 25.00 |
| | Trice | 156 | 70.91 |
| | 4 times | 5 | 2.27 |
| Which of the following lines of antiretroviral medications do you take? | First line | 194 | 88.18 |
| | Second line | 26 | 11.82 |
| For how long have you been taking it? | < 6 months | 15 | 6.82 |
| | 6 months—<12 months | 12 | 5.45 |
| | 1 –< 4 year | 50 | 22.73 |
| | 4 years and above | 143 | 65.00 |

60% of the participants were traders, with smaller proportions of artisans, unemployed, and public sector employees. Health-related behaviors included smoking (4.1%), alcohol consumption (76.4%), regular physical activity (26.8%), eating habits (70.9%). Antiretroviral medication usage [predominantly first-line (88.2%)], as well as ART durations which revealed that 65% of the participants have been on ART for at least 4 years.

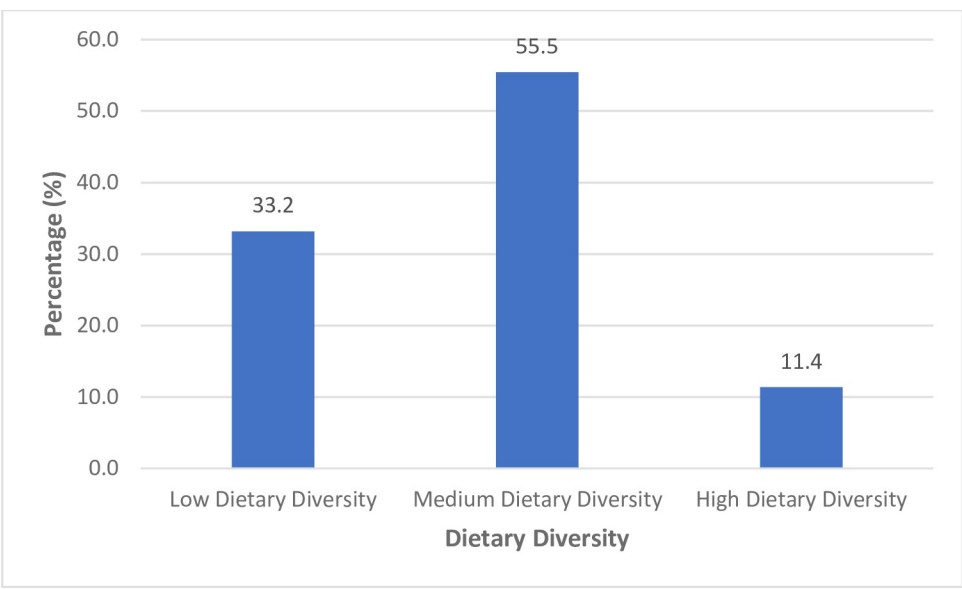

**Fig 1. Dietary diversity of participants.** The figure illustrates the distribution of dietary diversity among the study participants.

### Dietary diversity of participants

**Fig 1** presents results on the dietary diversity distribution among the study participants. A significant proportion (33.2%) of the study participants were identified as having low dietary diversity, with majority (55.5%) of the participants having medium dietary diversity (diet needing improvement).

### Nutritional status of participants

About 2 out of every 10 of the study participants were identified as being underweight (BMI <18.5 kg/m2). The results also show that less than half (43.5%) of all the participants had optimum nutritional status. **Fig 2** presents the distribution of the nutritional status of the study participants.

### Dietary diversity and the associated factors among study participants

**Table 2** presents results on the distribution and association of IDDS categories with the associated factors among the study participants. A significant association was found between the frequency of meal consumption in a day and the IDDS of the participants ($\chi 2$ = 19.219, p = 0.004, N = 220). Females were more evenly distributed, with 55% falling into the "medium" category, while males were predominantly in the "low" category (41.4%). However, no statistically significant association was observed between sex and IDDS, but occupation status of the study participants ($\chi 2$ = 28.650, p = 0.004, N = 220). A significant association was also observed between the line of antiretroviral medication (ART type) and length of antiretroviral use and IDDS ($\chi 2$ = 8.643, p = 0.013, N = 220; $\chi 2$ = 13.933, p = 0.030, N = 220, respectively).

### Nutritional status of study participants and associated factors

**Table 3** presents results on the distribution of nutritional status among participants based on demographic and health factors. It highlights associations between certain factors, such as sex,

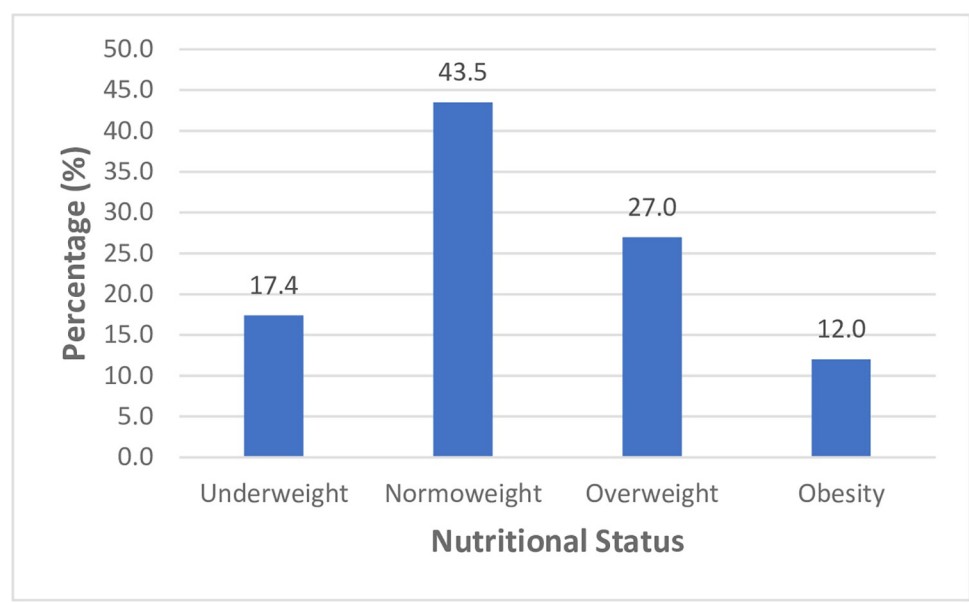

**Fig 2. Nutritional status of participants.** This figure presents the distribution of the nutritional status of the study participants.

education level, and occupation, and the participants' nutritional status. A significant association was observed between sex, occupation and nutritional status ($\chi 2 = 11.391$, p = 0.010, N = 220; $\chi 2 = 36.929$, p = 0.005, N = 220, respectively). Similarly, a significant association was found between educational level, ART type and nutritional status of the participants ($\chi 2 = 30.265$, p = 0.003, N = 220; $\chi 2 = 30.265$, p = 0.003, N = 220). Occupational categories exhibited varying distributions of nutritional status. Unemployed individuals had a higher proportion of individuals who were underweight (17.4%) compared to private and public sector workers. These findings may be helpful in understanding the relationships between socio-demographic characteristics and nutritional patterns within the study population.

## Predictors of dietary diversity

Study participants in the age category 40–59 years (reference: 20–39 years) were more likely to have a higher dietary diversity (AOR = 1.966, 95% CI: 1.045–4.987, p = 0.032). Study participants who ate at least three times daily (reference: at most twice daily) were more likely to have a high dietary diversity score (AOR = 1.641, 95% *CI*: 1.221, 8.879, p = 0.020). Study participants who had any form of employment were more likely to have a high dietary diversity compared to unemployed study participants. **Table 4** presents results on factors associated with the dietary diversity of the study participants. The final model exhibited a significant enhancement in fit compared to the null model [$\chi^2 = 55.432$, p < .001], suggesting its effectiveness. In total, the model explains 26.3% of the variability in IDDS (diet quality) (Nagelkerke pseudo R-square = 0.263).

## Predictors of nutritional status

Males were less likely to be underweight compared to females (AOR = 1.829, 95% CI: 1.294, 3.872, p = 0.007). Study participants taking ART second line medications were less likely to be underweight compared to participants who were on line one (AOR = 1.683, 95% CI: 1.282–2.424, P = 0.001). Similarly, the educational status categories; middle/JHS and Secondary when

**Table 2. Individual dietary diversity scores by associated factors Top of Form.**

| Factor | IDDS Category | | | Total, n (%) | Chi-Square | p-value |
|---|---|---|---|---|---|---|
| | Low, n (%) | Medium, n (%) | Higher, n (%) | | | |
| **Age Category** | | | | | | |
| 20–39 | 16 (44.4%) | 17 (47.2%) | 3 (8.3%) | 36 (100%) | 10.774 | 0.029 |
| 40–59 | 33 (24.8%) | 83 (62.4%) | 17 (12.8%) | 133 (100%) | | |
| 60+ | 24 (47.1%) | 22 (43.1%) | 5 (9.8%) | 51 (100%) | | |
| **Sex** | | | | | | |
| Female | 61(31.9%) | 105 (55%) | 25 (13.1%) | 191(100%) | 4.533 | 0.104 |
| Male | 12(41.4%) | 17 (58.6%) | 0 (0%) | 29 (100%) | | |
| **Marital Status** | | | | | | |
| Single | 17(25.0%) | 39 (57.4%) | 12 (17.6%) | 68 (100%) | 16.227 | 0.93 |
| Married | 14(32.6%) | 28 (65.1%) | 1 (2.3%) | 43 (100%) | | |
| Divorced | 3 (20%) | 12 (80%) | 0 (0%) | 15 (100%) | | |
| Widowed | 28(41.2%) | 31(45.6%) | 9(13.2%) | 68 (100%) | | |
| Separated | 1(33.3%) | 2(66.7%) | 0 (0%) | 3(100%) | | |
| Cohabiting | 10(43.5%) | 10(43.5%) | 3(13.0%) | 23(100%) | | |
| **Level of Education** | | | | | | |
| None | 19 (32.8%) | 36 (62.1%) | 3 (5.2%) | 58 (100%) | 7.169 | 0.518 |
| Primary | 14 (32%) | 25 (58.1%) | 4 (9.3%) | 43 (100%) | | |
| Middle/JHS | 31 (33%) | 47 (50.0%) | 16 (17.0%) | 94 (100%) | | |
| Secondary/SHS | 5 (33.3%) | 8 (53.3%) | 2 (13.3%) | 15 (100%) | | |
| Higher | 4 (40.0%) | 6 (60.0%) | 0 (0%) | 10 (100%) | | |
| **Occupation** | | | | | | |
| Unemployed | 11(50.0%) | 7 (31.8%) | 4 (18.2%) | 22 (100%) | 28.650 | 0.004 |
| Farmer | 9 (75.0%) | 3 (25.0%) | 0 (0%) | 12(100%) | | |
| Artisan | 10(35.7%) | 17(60.7%) | 1 (3.6%) | 28(100%) | | |
| Public Sector | 0(0%) | 12 (75.0%) | 4 (25.0%) | 16(100%) | | |
| Private Sector | 1 (20.0%) | 4 (80.0%) | 0 (0%) | 5(100%) | | |
| Trading | 42 (31.1%) | 77 (57.0%) | 16 (11.9%) | 135(100%) | | |
| Others | 0(0%) | 2 (100.0%) | 0 (0%) | 2(100%) | | |
| **Do you smoke cigarette?** | | | | | | |
| YES | 2 (22.2%) | 7 (77.8%) | 0(0%) | 9(100%) | 2.249 | 0.325 |
| NO | 71 (33.6%) | 115 (54.5%) | 25 (11.8%) | 211(100%) | | |
| **Do you drink alcohol?** | | | | | | |
| YES | 57 (33.9%) | 93 (55.4%) | 18 (10.7%) | 168(100%) | 0.384 | 0.825 |
| NO | 16 (30.8%) | 29 (55.8%) | 7 (13.5%) | 52(100%) | | |
| **Do you exercise?** | | | | | | |
| YES | 24 (40.7%) | 31 (52.5%) | 4 (6.8%) | 59 (100%) | 2.979 | 0.225 |
| NO | 49 (30.4%) | 91 (56.5%) | 21 (13.0%) | 161(100%) | | |
| **Do you eat outside home?** | | | | | | |
| No | 10 (47.6%) | 9 (42.9%) | 2 (9.5%) | 21 (100%) | 2.192 | 0.334 |
| Yes | 63 (31.7%) | 113 (56.8%) | 23 (11.6%) | 199(100%) | | |
| **How many times do you eat in a day?** | | | | | | |
| Once | 4 (100%) | 0(0%) | 0(0%) | 4(100%) | 19.219 | 0.004 |
| Twice | 18 (32.7%) | 25 (45.5%) | 12 (21.8%) | 55(100%) | | |
| Trice | 51 (32.7%) | 93 (59.6%) | 12 (7.7%) | 156(100%) | | |
| 4 times | 0 (0%) | 4 (80.0%) | 1 (20.0%) | 5(100%) | | |
| **Which of the following lines of antiretroviral medications do you take?** | | | | | | |

*(Continued)*

**Table 2.** (Continued)

| Factor | IDDS Category | | | Total, n (%) | Chi-Square | p-value |
|---|---|---|---|---|---|---|
| | Low, n (%) | Medium, n (%) | Higher, n (%) | | | |
| First line | 71 (36.6%) | 102 (52.6%) | 21 (10.8%) | 194(100%) | 8.643 | 0.013 |
| Second line | 2 (7.7%) | 20 (76.9%) | 4 (15.4%) | 26(100%) | | |
| **For how long have you been taking it?** | | | | | | |
| < 6 months | 2 (13.3%) | 12 (80.0%) | 1 (6.7%) | 15(100%) | 13.933 | 0.030 |
| 6 months—<12 months | 6 (50.0%) | 6 (50.0%) | 0(0%) | 12(100%) | | |
| 1 –< 4 year | 20 (40.0%) | 29 (58.0%) | 1 (2.0%) | 50(100%) | | |
| 4 years and above | 45 (31.5%) | 75 (52.4%) | 23 (16.1%) | 143(100%) | | |

compared to no formal educational attainment were less likely to be underweight (AOR = 1.211,95% CI: 1.002–5.321, AOR = 1.582, 95% CI: 1.318–2.948, respectively). Conversely, Participants that were on ART medication for at least one years were more likely to have higher body mass index (AOR = .181, 95% CI: 1.09–9.244, AOR = 1.837, 95% CI: 1.282–10.554, for 1 –< 4 year and 4 years and above, respectively). **Table 5** presents results on factors associated with the nutritional status of the study participants. There was a significant improvement in fit of the Final model over the null model [$\chi^2$ = 112.531, p < .001] indicating a good model. Overall, the model accounts for 42% of the variation in nutritional status *Negelkerke pseudo R square = 0.420*).

## Discussion

We conducted this study to investigate individual dietary diversity and nutritional status among PLHIV during the COVID-19 pandemic. The current study showed that majority of the participants were females with age category 50–59 years having the highest proportion of the age distribution. These findings are in sync with other published studies in Ghana which reported females to have a higher patronage of healthcare services in Ghana compared with males [20,21].

The results also indicate that majority (55.5%) of the study participants had medium dietary diversity (diet needing improvement) with a significant proportion (33.2%) actually having low dietary diversity (poor diet). Similar observations were made in a study conducted in Zambia, where a significant proportion of the study participants were found to have low dietary diversity [22]. Another study in Nigeria also reported high prevalent (42.7%) of low dietary diversity among PLHIV in their baseline findings [23]. This similarity underscores a broader regional challenge in maintaining diverse diets among PLHIV. The COVID-19 pandemic seemed to have a limited impact on the daily meal frequency among participants, as the vast majority of them reported consuming at least three meals in the last 24 hours, which aligns with the typical dietary pattern observed among Ghanaians [24]. However, the pandemic may have negatively affected their ability to ensure variety, as having frequent meal consumption does not necessarily translate into dietary diversity. The meals could actually be monotonous.

The COVID-19 pandemic possessed the capacity to significantly disrupt food security, affecting every aspect of food security, including production, supply, accessibility, availability, and utilization [25]. The government of Ghana implemented certain measures, which included lockdown and movement restrictions, as well as social distancing, in an effort to control the spread of COVID-19. These measures could have potentially impacted the availability and accessibility of various food groups [26], thereby resulting in low dietary diversity as observed in the current study. The study's identification of 33.2% of PLHIV as having low dietary

**Table 3. Nutritional status of study participants by associated factorsTop of Form.**

| Factor | Nutritional status | | | | Total, n (%) | Chi-Square | p-value |
|---|---|---|---|---|---|---|---|
| | Underweight | Normoweight | Overweight | Obese | | | |
| **Age Category** | | | | | | | |
| 20–39 | 9 (25.0%) | 15 (41.7%) | 7(19.4%) | 5 (13.9%) | 36 (100%) | 4.270 | 0.640 |
| 40–59 | 21 (15.1%) | 58 (41.7%) | 42 (30.2%) | 18 (12.9%) | 139 (100%) | | |
| 60+ | 10 (18.2%) | 27 (49.1%) | 13 (23.6%) | 5 (9.1%) | 55 (100%) | | |
| **Sex** | | | | | | | |
| Female | 38 (19.2%) | 79 (39.9%) | 53 (26.8%) | 28 (14.1%) | 198 (100%) | 11.391 | 0.010 |
| Male | 2 (6.3%) | 21 (65.6%) | 9 (28.1%) | 0 (0%) | 32 (100%) | | |
| **Marital Status** | | | | | | | |
| Single | 13 (19.1%) | 28 (41.2%) | 20 (29.4%) | 7 (10.3%) | 68 (100%) | 17.212 | 0.306 |
| Married | 6 (12.5% | 23 (47.9%) | 16 ()33.3% | 3 (6.3%) | 48 (100%) | | |
| Divorced | 4 (26.7%) | 9 (60.0%) | 2 (13.3%) | 0 (0%) | 15 (100%) | | |
| Widowed | 11 (15.5%) | 27 (38.0%) | 18 (25.4%) | 15 (21.1%)) | 71 (100%) | | |
| Separated | 2 (40.0%) | 3 (60.0%) | 0 (0%) | 0 (0%) | 5 (100%) | | |
| Cohabiting | 4 (17.4%) | 10 (43.5%) | 6 (26.1%) | 3 (13.0%) | 23 (100%) | | |
| **Level of Education** | | | | | | | |
| None | 15 (24.2%) | 27 (43.5%) | 18 (29.0%) | 2 (3.2%) | 62 (100%) | 30.265 | 0.003 |
| Primary | 13 (28.3%) | 18 (39.1%) | 12 (26.1%) | 3 (6.5%) | 46 (100%) | | |
| Middle/JHS | 12 (12.5%) | 41 (42.7%) | 24 (25.0%) | 19 (19.8%) | 96 (100%) | | |
| Secondary/SHS | 0 (0%) | 9 (60.0%) | 6 (40.0%) | 0 (0%) | 15 (100%) | | |
| Higher | 0 (0%) | 5 (45.5%) | 2 (18.2%) | 4 (36.4%) | 11 (100%) | | |
| **Occupation** | | | | | | | |
| Unemployed | 4 (17.4%) | 14 (60.9%) | 4 (17.4%) | 1 (4.3%) | 23 (100%) | 36.929 | 0.005 |
| Farmer | 4 (28.6%) | 8 (57.1%) | 2 (14.3%) | 0 (0%) | 28 (100%) | | |
| Artisan | 3 (10.7%) | 18 (64.3%) | 7 (25.0%) | 0 (0%) | 28 (100%) | | |
| Public Sector | 0 (0%) | 5 (31.3%) | 6 (37.5%) | 5 (31.3%) | 16 (100%) | | |
| Private Sector | 0 (0%) | 1 (20.0%) | 4 (80.0%) | 0 (0%) | 5 (100%) | | |
| Trading | 29 (20.4%) | 52 (36.6%) | 39 (27.5%) | 22 (15.5%) | 142 (100%) | | |
| Others | 0 (0%) | 2 (100%) | 0 (0%) | 0 (0%) | 2 (100%) | | |
| **Do you smoke cigarette?** | | | | | | | |
| YES | 2 (18.2%) | 5 (45.5%) | 4 (36.4%) | 0 (0%) | 11 (100%) | 1.800 | 0.615 |
| NO | 38 (17.4%) | 95 (43.4%) | 58 (26.5%) | 28 (12.8%)) | 219 (100%) | | |
| **Do you drink alcohol?** | | | | | | | |
| Yes | 31 (17.4%) | 77 (43.3%) | 48 (27.0%) | 22 (12.4%) | 178 (100%) | 0.031 | 0.999 |
| No | 9 (17.3%) | 23 (44.2%) | 14 (26.9%) | 6 (11.5%) | 52 (100%) | | |
| **Do you exercise?** | | | | | | | |
| YES | 7 (11.3%) | 26 (41.9%) | 23 (37.1%) | 6 (9.7%) | 62 (100%) | 5.535 | 0.137 |
| NO | 33 (19.6%) | 74 (44.0%) | 39 (23.2%) | 22 (13.1%) | 168 (100%) | | |
| **Do you eat outside home?** | | | | | | | |
| No | 6 (23.1%) | 12 (46.2%) | 5 (19.2%) | 3 (11.5%) | 26 (100%) | 1.252 | 0.741 |
| Yes | 34 (16.7%) | 88 (43.1%) | 57 (27.9%) | 25 (12.3%) | 204 (100%) | | |
| **How many times do you eat in a day?** | | | | | | | |
| Once | 2 (50.0%) | 2 (50.0%) | 0 (0%) | 0 (0%) | 4 (100%) | 15.055 | 0.089 |
| Twice | 9 (16.4%) | 16 (29.1%) | 20 (36.4%) | 10 (18.2%) | 55 (100%) | | |
| Thrice | 27 (16.3%) | 79 (47.6%) | 42 (25.3%) | 18 (10.8%) | 166 (100%) | | |
| 4 times | 2 (40.0%) | 3 (60.0%) | 0 (0%) | 0 (0%) | 5 (100%) | | |
| **Which of the following lines of antiretroviral medications do you take?** | | | | | | | |

*(Continued)*

**Table 3.** (Continued)

| Factor | Nutritional status | | | | Total, n (%) | Chi-Square | p-value |
|---|---|---|---|---|---|---|---|
| | Underweight | Normoweight | Overweight | Obese | | | |
| First line | 38 (18.7%) | 81 (39.9%) | 57 (28.1%) | 27 (13.3%) | 205 (100%) | 9.452 | 0.024 |
| Second line | 2 (7.4%) | 19 (70.4%) | 5 (18.5%) | 1 (3.7%) | 27 (100%) | | |
| **For how long have you been taking it?** | | | | | | | |
| < 6 months | 4 (23.5%) | 7 (41.2%) | 3 (17.6%) | 3 (17.6%) | 17 (100%) | 3.210 | 0.955 |
| 6 months—<12 months | 2 (16.7%) | 4 (33.3%) | 4 (33.3%) | 2 (16.7%)) | 12 (100%) | | |
| 1 –< 4 year | 8 (14.5%) | 23 (41.8%) | 16 (29.1%) | 8 (14.5%) | 55 (100%) | | |
| 4 years and above | 26 (17.8%) | 66 (45.2%) | 39 (26.7%) | 15 (10.3%) | 146 (100%) | | |

diversity is a matter of concern, especially in the context of a global health crisis like the COVID-19 pandemic. Dietary diversity is a critical component of a healthy and balanced diet as it ensures the intake of a variety of essential nutrients. A lack of dietary diversity can lead to nutritional deficiencies, which could further weaken the immune system and exacerbate health issues, particularly for PLHIV, who already face unique challenges related to their immune function.

Despite the progress made in mitigating weight and muscle loss among PLHIV with the advent of ART, underweight remains a persisting issue within this population. The findings from the present study indicate that a notable proportion (17.4%) of the study participants

**Table 4. Predictors of dietary diversity.**

| Characteristics | AOR | 95% CI | p-value |
|---|---|---|---|
| **Age** | | | |
| 20–39 | Ref | | |
| 40–59 | 1.966 | 1.045–4.987 | 0.032 |
| 60+ | 1.036 | .936–1.597 | 0.083 |
| **Sex** | | | |
| Female | Ref | | |
| Male | 0.882 | .029–3.832 | 0.657 |
| **Occupation** | | | |
| Unemployed | Ref | | |
| Farmer | 1.003 | .068–8.266 | 0.940 |
| Artisan | 1.076 | .079–14.655 | 0.956 |
| Public sector | 1.448 | 1.028–3.042 | 0.028 |
| Private sector | 1.165 | 1.030–9.329 | 0.041 |
| Trading | 1.403 | .103–9.035 | 0.799 |
| Others | 1.258 | .053–19.053 | 0.869 |
| **Do you exercise?** | | | |
| No | Ref | | |
| Yes | .965 | .478–1.948 | 0.921 |
| **Meal frequency per day** | | | |
| At most twice | Ref | | |
| At least thrice | 1.641 | 1.221–8.879 | 0.020 |
| **Which of the following lines of antiretroviral medications do you take?** | | | |
| First line | Ref | | |
| Second line | .598 | .278–10.948 | 0.971 |

**Table 5. Predictors of nutritional status.**

| Characteristics | AOR | 95% CI | p-value |
|---|---|---|---|
| **Sex** | | | |
| Female | Ref | | |
| Male | 1.829 | 1.294–3.872 | 0.007 |
| **Level of education** | | | |
| None | Ref | | |
| Primary | .651 | .078–1.948 | 0.215 |
| Middle/JHS | 1.211 | 1.002–5.321 | 0.021 |
| Secondary/SHS | 1.582 | 1.318–2.948 | 0.046 |
| Higher | 1.105 | .835–3.948 | 0.081 |
| **Occupation** | | | |
| Unemployed | Ref | | |
| Farmer | .813 | .008–1.966 | 0.113 |
| Artisan | 1.076 | .079–14.221 | 0.956 |
| Public sector | 1.048 | .028–7.402 | 0.568 |
| Private sector | 1.154 | .762–14.923 | 0.788 |
| Trading | 1.403 | 1.103–19.035 | 0.047 |
| Others | 1.258 | .083–19.053 | 0.869 |
| **Do you exercise?** | | | |
| No | Ref | | |
| Yes | .965 | .778–10.948 | 0.921 |
| **Meal frequency per day** | | | |
| At most twice | Ref | | |
| At least thrice | 1.441 | 1.021–1.879 | 0.020 |
| **Duration on ART** | | | |
| < 6 months | Ref | | |
| 6 months—<12 months | .833 | .082–3.124 | 0.621 |
| 1 –< 4 year | 1.181 | 1.09–9.244 | 0.032 |
| 4 years and above | 1.837 | 1.282–10.554 | 0.028 |

were classified as underweight, while those with normal weight comprised of less than 50% of the participants. These results is corroborated by a study in Senegal which reported the prevalence of undernutrition (BMI <18.5) to be 19.2% and 26.3% in Dakar and Ziguinchor respectively [27]. A more recent systematic review and meta-analysis that pooled the prevalence of undernutrition from 44 studies comprising 22,316 adult PLHIV in developing economies also reported the prevalence of undernutrition to be 23.72% [28]. This observation highlights the nuanced impact of COVID-19 on food security. The restrictive measures implemented by the government of Ghana, as noted earlier, likely contributed to the limited availability and accessibility of diverse food groups, intensifying the challenge of maintaining a nutritious diet among PLHIV.

Conversely, our findings indicate a higher prevalence compared to one other prior finding among women living with HIV in sub-Saharan Africa, which revealed a prevalence of 10.3% [29]. The study relied on secondary data from the demographic health and survey (DHS) and analyzed estimates from some selected sub-Saharan African countries. The disparity can potentially be attributed to changes in socio-demographic factors, HIV incidence trends, food security, and also but more importantly, COVID-19 pandemic. Additionally, it's important to note that the previous study exclusively focused on women, which could have influenced the outcome, even though the underlying biological mechanisms remain unclear.

PLHIV who fall within the age bracket of 40–59 years had more diverse diets compared with PLHIV within the age bracket of 20–39 years. This result is however at variance with previously published findings that reported a negative correlation between dietary diversity and aging [30,31]. The reason for the observation in this current study is probably because most of the working class among the PLHIV also fall within the age category of 40–59 years and so may afford more variety, and this may be explaining the current observation. Indeed, our findings show that PLHIV who are employed either in the public sector or the private sector had better dietary diversity as compared to PLHIV who are not employed. A Fijian study revealed that households with moderate dietary diversity (indicating a need for dietary improvement) exhibited a higher likelihood of unemployment [32].

Moreover, among PLHIV, those who consumed meals three times or more per day showed greater dietary diversity compared to those with a daily meal frequency of twice or less. This discovery aligns with a previous study that also found a positive association between increased meal frequency and higher dietary diversity [33]. This correlation is not surprising, as individuals who eat more frequently are more likely to incorporate a wider variety of foods into their diets.

Several factors demonstrated significant associations with undernutrition in our study, including sex, level of education, occupation, daily meal frequency, type of ART, and duration on ART. The findings revealed that female PLHIV were at a higher risk of experiencing undernutrition compared to their male counterparts, consistent with a prior study conducted in Ethiopia [34]. This gender disparity may stem from gender inequalities, cultural norms, restricted decision-making autonomy, as well as additional reproductive health and caregiving responsibilities assumed by women [35].

Regarding level of education among participants, our finding revealed that PLHIV who completed at least Junior High School were less prone to undernutrition. This is consistent with findings of a study conducted in Nepal which reported that literate PLHIV were less likely to be undernourished [36]. Also, PLHIV who had jobs, particularly traders, were less likely to be undernourished. A research conducted in Ethiopia suggests that occupation and economic status could potentially explain the heightened vulnerability to undernutrition among individuals with low levels of literacy [6]. Occupation and economic status were even worse as the world over experienced COVID-19 induced job losses and economic hardships.

Additionally, having three or more daily meal frequency was associated increased susceptibility for undernutrition among the study participants compared to PLHIV whose daily meal frequency was two or less. This outcome is possible because having a more frequent meals reduces the risk of overindulgence with regards to energy intake, as hunger does not get intense. The revelation is supported by one Korean study which reported that meal frequency was inversely associated to overnutrition [37]. The findings of this current study are however in contrast with a study conducted by Jordan P. et al., which revealed that an increase in meal frequency led to an increase in energy intake in all body mass index (BMI) categories [38], but they did not find a significant association between meal frequency and overweight.

Furthermore, the research revealed that PLHIV who had been on ART for a minimum of one year exhibited a reduced susceptibility to undernutrition, in line with previous studies [39,40]. This observation may be attributed to the immune system enhancement and lowered risk of opportunistic infections and nutritional deficiencies associated with prolonged ART usage [41].

The findings suggest that the final models are effective in explaining the variability in diet quality and nutritional status. Furthermore, the *Nagelkerke pseudo R-squared* values of 0.263 and 0.420 for the two models respectively indicate that they collectively account for 26.3% and 42% of the variation in diet quality and nutritional status respectively, which suggest that the

final models have meaningful predictive power and provide valuable insights into the factors influencing diet quality and nutritional status. The study however presented a few limitations. The utilization of a 24-hour recall method has inherent drawbacks. This method relies on a single day's dietary intake, which may not adequately characterize an individual's typical diet. As is the case with all retrospective data collection techniques, the primary limitation often associated with the 24-hour recall is its reliance on the subjects' ability to accurately remember and report what they consumed. This reliance on memory could potentially lead to either underestimation or overestimation of dietary intakes. Furthermore, the study evaluated the dietary diversity of participants using the IDDS established by the FAO. However, it is important to note that this scoring system does not capture detailed information regarding the quantities and specific nutrients consumed by the participants.

## Conclusion

The findings highlight the urgent need for targeted interventions to enhance dietary diversity and nutritional well-being among PLHIV, especially during pandemics when traditional food security measures may falter. Healthcare providers, such as nutritionists and dietitians, should develop and implement tailored nutritional programs for PLHIV, emphasizing diverse and balanced diets, overcoming barriers to nutritious food access, and integrating regular nutritional assessments. These measures address immediate nutritional deficiencies and improve overall health and immune function, potentially mitigating the adverse effects of HIV and COVID-19.

Factors like age, sex, occupation, exercise, meal frequency, ART duration, and treatment regimen significantly affect dietary diversity and undernutrition. Individuals with these characteristics should adopt balanced diets, incorporate regular physical activity, and seek personalized guidance from healthcare professionals for healthy weight management.

On a policy and research level, collaboration among government agencies, NGOs, and research institutions is crucial to investigate the causes of limited dietary diversity and high undernutrition rates among PLHIV during crises. Policies should prioritize improving food security and accessibility for marginalized groups, such as PLHIV, and be informed by evidence-based research. Inclusive food assistance programs and investment in training healthcare professionals, particularly dietitians and nutritionists specializing in HIV-specific nutritional care, are essential. Future studies should examine the long-term impacts of improved dietary diversity on PLHIV health outcomes and the role of socioeconomic, cultural, and environmental factors in shaping dietary habits. By addressing these aspects, comprehensive strategies can be developed to improve the nutritional status of PLHIV and enhance their resilience against future public health emergencies.

## Supporting information

**S1 Data.**
(CSV)

## Acknowledgments

We express our sincere gratitude to the PLHIV attending Cape Coast Teaching Hospital and University of Cape Coast Hospital for their willingness to participate in our study. Our thanks also extend to the management boards as well as staff of the ART clinics for granting us access and facilitating our data collection efforts. Additionally, we are appreciative of the research assistants who were enlisted to assist in gathering data.

## Author Contributions

**Conceptualization:** Kasim Abdulai, Safianu Osman Aleboko, Mohammed Doobia Ahmed, Odei-Asare Fremah Adom, Rhoda Kumah.

**Data curation:** Kasim Abdulai, Abdul Rauf Alhassan, Odei-Asare Fremah Adom.

**Formal analysis:** Kasim Abdulai, Abdul Rauf Alhassan, Safianu Osman Aleboko.

**Investigation:** Kasim Abdulai.

**Methodology:** Kasim Abdulai, Safianu Osman Aleboko, Mohammed Doobia Ahmed, Awal Seidu Mohammed.

**Project administration:** Kasim Abdulai.

**Resources:** Kasim Abdulai.

**Supervision:** Kasim Abdulai, Awal Seidu Mohammed.

**Writing – original draft:** Kasim Abdulai, Abdul Rauf Alhassan, Safianu Osman Aleboko, Mohammed Doobia Ahmed, Awal Seidu Mohammed, Odei-Asare Fremah Adom, Rhoda Kumah.

**Writing – review & editing:** Kasim Abdulai, Safianu Osman Aleboko, Mohammed Doobia Ahmed, Awal Seidu Mohammed, Odei-Asare Fremah Adom, Rhoda Kumah.

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
