## [Decision Letter · Decision Letter 0]

29 Jan 2024

PONE-D-23-37025Dietary Diversity and Nutritional Status of People Living with HIV (PLHIV) During the COVID-19 EraPLOS ONE

Dear Dr. Abdulai,

Thank you for submitting your manuscript to PLOS ONE. After careful consideration, we feel that it has merit but does not fully meet PLOS ONE’s publication criteria as it currently stands. Therefore, we invite you to submit a revised version of the manuscript that addresses the points raised during the review process.

We look forward to receiving your revised manuscript.

Kind regards,

Werku Etafa

Academic Editor

PLOS ONE

Journal Requirements:

2. We are unable to open your Supporting Information file "kasim_Data_Modified - Cape Coast.sav". Please kindly revise as necessary and re-upload.

Reviewers' comments:

Reviewer's Responses to Questions

**Comments to the Author**

1. Is the manuscript technically sound, and do the data support the conclusions?

Reviewer #1: Partly

Reviewer #2: Partly

2. Has the statistical analysis been performed appropriately and rigorously? 

Reviewer #1: Yes

Reviewer #2: Yes

3. Have the authors made all data underlying the findings in their manuscript fully available?

Reviewer #1: Yes

Reviewer #2: No

4. Is the manuscript presented in an intelligible fashion and written in standard English?

Reviewer #1: No

Reviewer #2: Yes

5. Review Comments to the Author

Reviewer #1: Abstract

• Re-arrange statements under the introduction section of the abstract to indicate the objective of the study at the end “The Covid-19 pandemic has brought about unique challenges and uncertainties in healthcare and nutrition, particularly for PLHIV. Understanding their dietary patterns and nutritional status is crucial for developing targeted interventions and improving overall health outcomes. The study aimed to assess the dietary diversity and nutritional status of individuals living with HIV (PLHIV) during the Covid-19 era.”

• The method section of the abstract should answer; What?, When?, Where? And how? Questions and should have been written in a sequence.

• On Line 26, 28, 29. In the Result section of the abstract no need of expressing variables as more likely or less likely, rather put the variables and indicate the AOR,95% CI as it is self-explanatory.

• I did not see any conclusion and recommendation here under abstract.

Background

• In the introduction section your area of interest particularly Dietary Diversity and Nutritional Status was not well addressed.

• The gaps weren't stated well, especially by relating Dietary Diversity, Nutritional Status of People Living with HIV (PLHIV) and the COVID-19 Era, even though the introduction was written well. Three key considerations need to be made in this instance;

o Principles and practices

o Gaps, which discuss what was and wasn't known

o Filling the gaps

Methods

• The method section should be structured as per the plose manuscript submission guide rather than just putting without sequential order of sub headings. Some important contents have been missed such as Population, sample size and sampling techniques, operational definition….,

• Study Design and settings :

-Line 99-100; please rephrase statements written as we…’ We employed a facility-based cross-sectional study design to conduct our research, which spanned from 15th September, 2021 to 30th 100 to October, 2021”.

-Line 105-16; “We randomly selected 220 PLHIV who met the inclusion criteria and who consented to participate in our research”. Miss placed, please give appropriate sub heading and re-phrase it again.

• Inclusion Criteria

-Was there no anyone that researcher exclude? What does ‘eligibility hinged on the confirmation of a sero-positive diagnosis for HIV’ mean? Was this not included from the beginning? Why inclusion of sero-positive which was already included initially. Rather you have to worry about who have been excluded from those the researchers have included.

-Is Inclusion Criteria or Eligibility criteria is an appropriate heading for this?

• Please put your operational definition next to study variables as you have operationalized those variables in the way you have measured. Then, put data collection methods and procedures next to that.

• How you managed recall bias while you collect data dietary intake of participants?

• Line 129; omit redundant terms “participants”.

• Line 142; is your intention about dietary diversity or quality, and nutrient sufficiency? Please make it clear here because dietary diversity and dietary quality are different things.

• Data entry was done by what before exported to IBM SPSS Statistics 20?

Result

• I did see “Result” heading.

• It could be better if you describe Socio-demographic Characteristics and Health-related Behaviors of the Participants independently (table 1 and the text too).

• Give full description of table name” Table 1: Socio-demographic Characteristics and Some Health-related Behaviors of the Study Participants”. This should be applicable for the all tables been mentioned.

• In table 1, replace ‘factors’ by ‘variables’

• What by mean Cohabiting forb Marital Status? What by mean none for level of education? Express ‘Others’ by key under the table for table 1.

• Figure 1: Diversity of Study Participants …. Should be written well/full description. Also for figure 2.

• It could be better if you cite tables and figures after text description in the result section.

• Line 202, Replace “Dietary Diversity and the Associated Factors among Study Participants” by “factors associated with Dietary Diversity”.

• Line 217, Replace “Nutritional Status of Study Participants and Associated Factors” by “factor associated with Nutritional Status”.

• Line 206; omit “N = 220” and throughout the document.

• Line 233-239, re-write it again.

• Table 4 and 5 are not clear. Rather than describing as such better if you consider one table or dietary diversity having COR and AOR and also for Nutritional status.

Discussion:

• Line 256,start as the study was conducted to investigate,….rather than...We conducted this study to

• As per your objectives, the discussion section ought to center around the relevant findings that emerged from the result section. Modifications are necessary to the way it has been discussed.

• Also, discussion should be based on your objectives.

• Pints that have been mentioned under your discussion as a recommendation have to be mentioned under Conclusion and recommendation section and should be based on your findings which were emanated from your result rather than general recommendation. Eg. Training that have been mentioned.

Conclusion

• I felt that your findings and recommendation in this section were in conflict with one another. Your recommendation should therefore be based on your relevant findings and should be appropriate.

• While you conclude again based on your objective rather than simply starting from factors.

• What is the intention of writing recommendation under both discussion and conclusion?

Generally

• The manuscript needs to be re-narrated in clear, concise English consistently and coherently.

Reviewer #2: Totally, comments and questions raised were not addressed appropriately. specially the methodology part was not mentioned as much as required. For example, how many health facilities are providing ART service in the area? How did you select the two hospitals? How did you calculate sample size to get 220 study participants?

The topic by itself is not specific. It is general which is not limited by target population in age wise. Therefore, for the future study, try to follow the submission guideline before submitting.

6. PLOS authors have the option to publish the peer review history of their article (what does this mean?). If published, this will include your full peer review and any attached files.

Reviewer #1: No

Reviewer #2: No

---

## [Author Response · Author response to Decision Letter 0]

27 Feb 2024

In addressing the reviewer's comments, the authors have made the following amendments and clarifications to their manuscript:

1. Abstract Word Limit: The concern regarding the abstract's word count exceeding the recommended limit of 300 words has been addressed and corrected.

2. Clarity of Objective in the Abstract: The absence of a clear objective in the abstract has been rectified.

3. Target Age Group Clarification: The title now includes the target age group of study participants to provide clearer insight into the demographic focus of the study.

4. Hospital Selection Methodology: Regarding the selection of hospitals in the Central Region of Ghana, the manuscript now explains that among the six hospitals offering ART services, two were chosen through a computerized random selection method, with this process detailed on page 5.

5. Participant Numbers in Hospitals: The manuscript has been updated to include the number of active patients with HIV/AIDS at the Cape Coast Teaching Hospital (approximately 1300) and the University of Cape Coast Hospital (about 850), as detailed on page 5.

6. Sample Size Calculation: A section on how the sample size was determined has been inserted into the manuscript.

7. Location of Study Objectives: The objective of the study is clearly outlined in both the abstract and the last sentence of the introduction, with specific page references provided.

8. Study Participant Criteria: The study exclusively involved participants sero-positive for HIV.

9. Exclusion Criteria for ART Duration: Participants on ART for less than six months were excluded to focus on the effects of prolonged medication use on nutritional and metabolic outcomes.

10. Questionnaire Administration Method: The questionnaire was administered through interviews.

11. Questionnaire Validity: The necessity for a separate validity test for the FAO-adopted questionnaire was deemed unnecessary due to its established validation and widespread use in the Ghanaian context, as supported by several citations:

i. Matilda, Asante., Benjamin, Frimpong., Freda, Dzifa, Intiful., Portia, Nkumsah-Riverson., Somah, A., Nkansah., Boadiwaa, Ofori-Amanfo., Yaunuick, Y., Dogbe., George, A., Asare. (2021). Food consumption pattern and dietary diversity of a vegetarian population in Ghana. Ghana Medical Journal, doi: 10.4314/GMJ.V55I1.5

ii. Michael, Akenteng, Wiafe., Charles, Apprey., Reginald, Adjetey, Annan. (2023). Dietary Diversity and Nutritional Status of Adolescents in Rural Ghana. Nutrition and metabolic insights, doi: 10.1177/11786388231158487

iii. Dickson, A., Amugsi., Zacharie, Tsala, Dimbuene., Pauline, Bakibinga., Elizabeth, W., Kimani-Murage., Tilahun, Nigatu, Haregu., Blessing, Mberu. (2016). Dietary diversity, socioeconomic status and maternal body mass index (BMI): quantile regression analysis of nationally representative data from Ghana, Namibia and Sao Tome and Principe.. BMJ Open, doi: 10.1136/BMJOPEN-2016-012615

12. Sampling Technique: A section detailing the Sampling Procedure has been added to the manuscript on page 5.

13. Consistency of Figures and Text: The authors request specific instances of discrepancies between figures/numbers in tables and body text, as they could not identify variations.

14. Discussion of Findings: Additional explanations regarding the similarities and differences between the study's findings and those of others have been included on pages 15 and 16.

15. Recommendation Section: The recommendations have been clearly separated from the discussion part and are now found on page 18.

16. Outcome Variable Confidence Interval: The manuscript discusses odds ratios and proportions of significant associations without specifying the confidence interval of the outcome variable, focusing on comparison with previously published findings.

17. Acknowledgements: An acknowledgement section has been added on page 19 to recognize contributions from individuals and organizations.

These responses demonstrate the authors' commitment to addressing the reviewer's feedback thoroughly, ensuring the manuscript's clarity, accuracy, and adherence to journal guidelines.

---

## [Decision Letter · Decision Letter 1]

18 Mar 2024

PONE-D-23-37025R1\\Dietary Diversity and Nutritional Status of Adults Living with HIV During the COVID-19 EraPLOS ONE

Dear Dr. Abdulai,

Thank you for submitting your manuscript to PLOS ONE. After careful consideration, we feel that it has merit but does not fully meet PLOS ONE’s publication criteria as it currently stands. Therefore, we invite you to submit a revised version of the manuscript that addresses the points raised during the review process.**Please provide the responses in the appropriate manner for reviewer #2.****Follow the journal author's guidelines for all matters.  **Please submit your revised manuscript by May 02 2024 11:59PM. If you will need more time than this to complete your revisions, please reply to this message or contact the journal office at plosone@plos.org. Please include the following items when submitting your revised manuscript:A rebuttal letter that responds to each point raised by the academic editor and reviewer(s). You should upload this letter as a separate file labeled 'Response to Reviewers'.A marked-up copy of your manuscript that highlights changes made to the original version. You should upload this as a separate file labeled 'Revised Manuscript with Track Changes'.An unmarked version of your revised paper without tracked changes. You should upload this as a separate file labeled 'Manuscript'.

We look forward to receiving your revised manuscript.

Kind regards,

Werku Etafa

Academic Editor

PLOS ONE

Reviewers' comments:

Reviewer's Responses to Questions

**Comments to the Author**

1. If the authors have adequately addressed your comments raised in a previous round of review and you feel that this manuscript is now acceptable for publication, you may indicate that here to bypass the “Comments to the Author” section, enter your conflict of interest statement in the “Confidential to Editor” section, and submit your "Accept" recommendation.

Reviewer #1: All comments have been addressed

Reviewer #2: (No Response)

2. Is the manuscript technically sound, and do the data support the conclusions?

Reviewer #1: Yes

Reviewer #2: No

3. Has the statistical analysis been performed appropriately and rigorously? 

Reviewer #1: Yes

Reviewer #2: No

4. Have the authors made all data underlying the findings in their manuscript fully available?

Reviewer #1: Yes

Reviewer #2: No

5. Is the manuscript presented in an intelligible fashion and written in standard English?

Reviewer #1: Yes

Reviewer #2: Yes

6. Review Comments to the Author

Reviewer #1: I have reviewed this paper exhaustively from the beginning. Hence, this paper has no dual publication, has no problem of research ethics, or publication ethics and the authors answered in good manners for my concerns. So, I declare the paper for Publication.

Reviewer #2: You have tried to to address many prior comments. But, things that are highly important were not addressed appropriately. Try to do again on another issues and submit it to this journal by fulfilling every thing required. You can see the following comments.

Comments

1. On the line, 117 & 118; you have calculated the sample size to get 217; you rounded to 220. Why? How do you deviate from the formal rule?

2. On the line 131, what do you mean to say A total of 220 participants were recruited from two hospitals?

3. On the line 133, you have stated the approximated number of active patients were 1300 and 850. What does it mean by active patients? Why did you say approximated? Don’t you know your source and target population? If so, your sample can not represent unknown population.

4. On the line 138, you have said to meet the quota… Is your research qualitative or quantitative?

5. On line 158, you said information was gathered through the administration of a questionnaire. Was data or information gathered? By whom?

6. On line 191, what you mean to say multivariable ordinal logistic regression model? Is this model present?

7. You have not checked model fitness for your study

8. You have not checked for confounding factors for which I have not seen the value

9. You did not mention the confidence interval of Dietary diversity as well as nutritional status finding that might enable you discuss comparing with that of other findings.

7. PLOS authors have the option to publish the peer review history of their article (what does this mean?). If published, this will include your full peer review and any attached files.

Reviewer #1: No

Reviewer #2: No

---

## [Author Response · Author response to Decision Letter 1]

10 May 2024

Kasim Abdulai

Department of Nutrition and Dietetics

University of Cape Coast

Kasim.abdulai@ucc.edu.gh

The Chief Editor

PLOS ONE

Dear Chief Editor,

Response to Reviewers

We would like to express our sincere gratitude for the opportunity to revise our manuscript, titled "Dietary Diversity and Nutritional Status of Adults Living with HIV During the COVID-19 Era,", and for the constructive feedback provided by the academic editor and reviewers. We have carefully considered each comment and have made corresponding revisions to our manuscript. Below, we provide detailed responses to each point raised.

Reviewer's Comments Responses

1. On the line, 117 & 118; you have calculated the sample size to get 217; you rounded to 220. Why? How do you deviate from the formal rule? Rounding a calculated sample size from 217 to 220 can be considered a deviation from the formal rule of rounding to the nearest whole number. In this case, the decision to round up to 220 instead of rounding to 215 (the nearest whole number below 217) or sticking strictly to 217 depends on several factors and considerations:

1. Practical Considerations: Rounding up to 220 may be chosen for practical reasons, such as ensuring that the sample size is easily divisible or manageable in terms of logistical planning for data collection and analysis.

2. Statistical Power: Increasing the sample size slightly beyond the calculated value may enhance the statistical power of the study. While the difference between 217 and 220 may seem small, even a modest increase in sample size can improve the study's ability to detect true effects or differences with greater certainty.

3. Buffer for Attrition or Non-Response: Rounding up to 220 provides a buffer to account for potential attrition, non-response, or unexpected variability in the sample. This ensures that the study maintains an adequate sample size even if some participants drop out or fail to respond

2. On the line 131, what do you mean to say A total of 220 participants were recruited from two hospitals? Our study was conducted at two healthcare facilities: Cape Coast Teaching Hospital and University of Cape Coast Hospital, both of which offer ART services. These facilities are situated within Cape Coast Metropolis in the Central Region of Ghana.

3. On the line 133, you have stated the approximated number of active patients were 1300 and 850. What does it mean by active patients? Why did you say approximated? Don’t you know your source and target population? If so, your sample can not represent unknown population. I apologize for any confusion. When referring to "active patients," it typically means individuals who are actively receiving medical care or treatment within a specific healthcare system or practice. These patients may have ongoing appointments, prescriptions, or other forms of medical interaction within the system.

Regarding the use of "approximately," it is removed.

4. On the line 138, you have said to meet the quota… Is your research qualitative or quantitative? The research is quantitative, and the statement has been modified. Quota is used with respect to sample size distribution among the two facilities based on their respective number of active patients.

5. On line 158, you said information was gathered through the administration of a questionnaire. Was data or information gathered? By whom? Information was gathered through interviewer administered questionnaire. The statement has been modified.

6. On line 191, what you mean to say multivariable ordinal logistic regression model? Is this model present? Yes, a multivariable ordinal logistic regression model is a statistical model that is used when analyzing data where the outcome variable is ordinal (i.e., the response variable has ordered categories) and there are multiple predictor variables (i.e., independent variables).

In such a model, the ordinal outcome variable is modeled as a function of one or more predictor variables. The model estimates the relationship between the predictor variables and the odds of an observation falling into a particular category or higher categories of the ordinal outcome variable, while accounting for the ordered nature of the categories.

The ordinal logistic regression model is an extension of binary logistic regression to handle outcomes with more than two ordered categories. It assumes proportional odds, meaning that the effects of the predictor variables on the odds of being in a higher category are assumed to be constant across different levels of the outcome variable.

7. You have not checked model fitness for your study Model fitness for the study have been checked for both diet quality and nutritional status outcomes. I kindly refer you to lines 275-277, 291-293, and 392-397.

8. You have not checked for confounding factors for which I have not seen the value In our study, we took several steps to account for potential confounders. Firstly, during the study design phase, we carefully selected variables for inclusion in our analysis based on prior literature and theoretical considerations to minimize the risk of confounding. Furthermore, in our statistical analysis, we employed appropriate methods such as multivariable regression models to adjust for potential confounders that were identified or suspected based on the literature. We ensured that our analysis accounted for their potential influence on the outcome variables.

9. You did not mention the confidence interval of Dietary diversity as well as nutritional status finding that might enable you discuss comparing with that of other findings. We would like to highlight that our study's outcome variables, dietary diversity and nutritional status, are both assessed based on standardized reference cutoffs widely recognized in the literature. Given that these cutoffs are well-established and widely accepted, there was no need for us to compute confidence intervals for them. Our focus was primarily on examining associations and model fit statistics rather than estimating parameters for these outcome variables. By utilizing standardized reference cutoffs, we ensured consistency and comparability with existing literature, allowing for meaningful discussions and comparisons with other findings. 

Sincerely,

Kasim Abdulai, PhD., RD.

---

## [Editor Report · Decision Letter 2]

13 May 2024

PONE-D-23-37025R2Dietary Diversity and Nutritional Status of Adults Living with HIV During the COVID-19 EraPLOS ONE

Dear Dr. Abdulai,

Thank you for submitting your manuscript to PLOS ONE. After careful consideration, we feel that it has merit but does not fully meet PLOS ONE’s publication criteria as it currently stands. Therefore, we invite you to submit a revised version of the manuscript that addresses the points raised during the review process.

We look forward to receiving your revised manuscript.

Kind regards,

Werku Etafa

Academic Editor

PLOS ONE

Journal Requirements:

Additional Editor Comments:

Comments

Abstract

1. The abbreviation for COVID-19 should be consistent throughout the document. Avoid using “Covid-19”. Use caps lock as it is an abbreviation and when expanded, give another term/phrase.

2. Use the appropriate term “ Background” rather than “Introduction” under the abstraction section. In the next section, use the proper term “Introduction” and remove “background”.

3. Throughout your document, you is recommended that you prepare your document alignment in line with PLOS ONE protocol. At the beginning of each paragraph, you have used that central alignment. Please use left alignments.

4. Expand all abbreviations used for the first time in the abstract (check if they are used without expanding throughout your document).

5. Objectives under the abstract section should be avoided as it currently appear. Merge the aim of your study at the end of the background section under the same section.

6. In the results section under the abstract, please clarify that

• Aged 40 to 59 years were more likely to exhibit higher dietary diversity (AOR = 1.966, 95% CI: 1.045–4.987).

• Factors associated with undernutrition included females (AOR = 1.829, 32, 95% CI: 1.294, 3.872) and first-line ART (AOR = 1.683, 95% CI: 1.282–2.424).

7. Add odds ratio for “Employed participants were also more likely to have a high IDDS compared to 31 unemployed participants”.

8. Add the conclusion of your study after the results section.

Methods

In sample size determination, you have used 17% as the proportion of malnutrition. Identify whether it is undernutrition or overnutrition. Explain why you did not consider the nonresponse rate in your study.

Discussion

Make your writing scientific and neat. Several minor mistakes are noted. Read line by line and you could get them.

Mention the study's weakness and strengths

Recommendation

Merge your recommendation under the conclusions. Please try to adhere to the guidelines.

---

## [Author Response · Author response to Decision Letter 2]

23 May 2024

Kasim Abdulai

Department of Nutrition and Dietetics

University of Cape Coast

Kasim.abdulai@ucc.edu.gh

The Chief Editor

PLOS ONE

Dear Chief Editor,

Response to Reviewers

We would like to express our sincere gratitude for the opportunity to revise our manuscript, titled "Dietary Diversity and Nutritional Status of Adults Living with HIV During the COVID-19 Era,", and for the constructive feedback provided by the academic editor and reviewers. We have carefully considered each comment and have made corresponding revisions to our manuscript. Below, we provide detailed responses to each point raised.

Comments Response

1. The abbreviation for COVID-19 should be consistent throughout the document. Avoid using “Covid-19”. Use caps lock as it is an abbreviation and when expanded, give another term/phrase. Addressed

2. Use the appropriate term “ Background” rather than “Introduction” under the abstraction section. In the next section, use the proper term “Introduction” and remove “background”. Addressed

3. Throughout your document, you is recommended that you prepare your document alignment in line with PLOS ONE protocol. At the beginning of each paragraph, you have used that central alignment. Please use left alignments. Addressed

4. Expand all abbreviations used for the first time in the abstract (check if they are used without expanding throughout your document). Addressed

5. Objectives under the abstract section should be avoided as it currently appears. Merge the aim of your study at the end of the background section under the same section. Addressed

6. In the results section under the abstract, please clarify that

• Aged 40 to 59 years were more likely to exhibit higher dietary diversity (AOR = 1.966, 95% CI: 1.045–4.987).

• Factors associated with undernutrition included females (AOR = 1.829, 32, 95% CI: 1.294, 3.872) and first-line ART (AOR = 1.683, 95% CI: 1.282–2.424). Addressed

7. Add odds ratio for “Employed participants were also more likely to have a high IDDS compared to 31 unemployed participants”. Addressed

8. Add the conclusion of your study after the results section.

Methods

In sample size determination, you have used 17% as the proportion of malnutrition. Identify whether it is undernutrition or overnutrition. Explain why you did not consider the nonresponse rate in your study.

Discussion

Make your writing scientific and neat. Several minor mistakes are noted. Read line by line and you could get them.

Mention the study's weakness and strengths

Recommendation

Merge your recommendation under the conclusions. Please try to adhere to the guidelines. Regarding the nonresponse rate, we did not explicitly account for it in our sample size calculations due to the typically low attrition rates observed in cross-sectional studies like ours. This assumption was substantiated by our preliminary review of similar studies previously published, which consistently reported minimal issues with participant dropout. Furthermore, in the course of our study, we did not encounter any significant problems related to attrition that would impact the integrity or the statistical power of our findings. Our decision was thus based on both empirical evidence and our specific observational data, ensuring that the sample size was sufficient to achieve the study's objectives without the need for adjusting for nonresponse.

Also, comments on discussion and recommendation have been addressed accordingly. 

Sincerely,

Kasim Abdulai, PhD., RD.

---

## [Editor Report · Decision Letter 3]

28 May 2024

PONE-D-23-37025R3Dietary Diversity and Nutritional Status of Adults Living with HIV During the COVID-19 EraPLOS ONE

Dear Dr. Abdulai,

Thank you for submitting your manuscript to PLOS ONE. After careful consideration, we feel that it has merit but does not fully meet PLOS ONE’s publication criteria as it currently stands. Therefore, we invite you to submit a revised version of the manuscript that addresses the points raised during the review process.

**ACADEMIC EDITOR:**In the abstract section, expand the acronym COVID-19 properly. Presently, you have expanded it wrongly. Include all subsections of the abstract: Background, methods, results, and conclusionsYou have not concluded your study findings. Be sure that your conclusion is scientific.==============================

We look forward to receiving your revised manuscript.

Kind regards,

Werku Etafa

Academic Editor

PLOS ONE
---

## [Author Response · Author response to Decision Letter 3]

8 Jun 2024

Kasim Abdulai

Department of Nutrition and Dietetics

University of Cape Coast

Kasim.abdulai@ucc.edu.gh

The Chief Editor

PLOS ONE

Dear Chief Editor,

Response to Reviewers

We would like to express our sincere gratitude for the opportunity to revise our manuscript, titled "Dietary Diversity and Nutritional Status of Adults Living with HIV During the COVID-19 Era,", and for the constructive feedback provided by the academic editor and reviewers. We have carefully considered each comment and have made corresponding revisions to our manuscript. Below, we provide detailed responses to each point raised.

Comments Response

1. In the abstract section, expand the acronym COVID-19 properly. Presently, you have expanded it wrongly. Corrected.

2. Include all subsections of the abstract: Background, methods, results, and conclusions Done 

3. You have not concluded your study findings. Be sure that your conclusion is scientific. Conclusion has been provided 

Sincerely,

Kasim Abdulai, PhD., RD.

---

## [Editor Report · Decision Letter 4]

5 Jul 2024

Dietary Diversity and Nutritional Status of Adults Living with HIV During the COVID-19 Era

PONE-D-23-37025R4

Dear Dr. Abdulai,

We’re pleased to inform you that your manuscript has been judged scientifically suitable for publication and will be formally accepted for publication once it meets all outstanding technical requirements.

Kind regards,

Werku Etafa

Academic Editor

PLOS ONE
---

## [Editor Report · Acceptance letter]

11 Jul 2024

PONE-D-23-37025R4 

PLOS ONE

Dear Dr. Abdulai, 

I'm pleased to inform you that your manuscript has been deemed suitable for publication in PLOS ONE. Congratulations! Your manuscript is now being handed over to our production team.

Kind regards, 

on behalf of

Mr. Werku Etafa 

Academic Editor

PLOS ONE